# Evaluation on Context Recognition Using Temperature Sensors in the Nostrils [note 1]

**DOI:** 10.3390/s19071528

**Published:** 2019-03-29

**Authors:** Ryosuke Kodama, Tsutomu Terada, Masahiko Tsukamoto

**Affiliations:** 1Graduate School of Engineering, Kobe University, 1-1 Rokkodaicho, Nada, Kobe, Hyogo 657-8501, Japan; tsutomu@eedept.kobe-u.ac.jp (T.T.); tuka@kobe-u.ac.jp (M.T.); 2Strategic Creation Research Promotion Project (PRESTO) of the Japan Science and Technology Agency (JST), 4-1-8 Honmachi, Kawaguchi, Saitama 332-0012, Japan

**Keywords:** wearable computing, context recognition, nasal cycle, temperature sensor

## Abstract

We can benefit from various services with context recognition using wearable sensors. In this study, we focus on the contexts acquired from sensor data in the nostrils. Nostrils can provide various contexts on breathing, nasal congestion, and higher level contexts including psychological and health states. In this paper, we propose a context recognition method using the information in the nostril. We develop a system to acquire the temperature in the nostrils using small temperature sensors connected to glasses. As a result of the evaluations, the proposed system can detect breathing correctly, workload at an accuracy of 96.4%, six behaviors at an accuracy of 54%, and eight behaviors in daily life at an accuracy of 86%. Moreover, the proposed system can detect nasal congestion, therefore, it can log nasal cycles that are considered to have a relationship with the autonomic nerves and/or biological states.

## 1. Introduction

People can benefit from various services provided by wearable sensors for context recognition. For example, lifelog services have become popular, which is a chronicle consisting of data such as position history from GPS or pictures taken by camera in a person’s daily life. Also, the data is recorded automatically by wearable devices. In recent years, we have been sometimes able to take care of our health or detect a health problem by logging data such as activity or sleep state using wearable sensors in our daily lives. Such information can be utilized to provide appropriate feedback to keep the user healthy. Recently, acquired contexts from wearable sensors include not only physical contexts such as user motion and position but also psychological contexts such as stress.

In this research, we focus on data from the nose among the data that can be acquired with wearable devices. Among them, we think that breathing and nasal congestion include rich information. Breathing contains psychological and biological information. Although there are many wearable devices that can detect breath, they restrict users to a specific environment or cannot detect breath with high accuracy. Nasal congestion bothers many people, however, its causes are complicated and remain unknown. If we know the conditions in the nostrils for various situations and long time, we can determine the mechanism of the nasal cycle or people suffering from nasal congestion can measure their tendency to nasal congestion with a wearable device. Moreover, there is a biological phenomenon called a nasal cycle associated with the alternation of congestion and decongestion in the right and left nostrils. The nasal cycle remains a subject of debate, and it is often argued that the nasal cycle has a relationship with the activity of a cerebral hemisphere [1]. There is a strong possibility to discover/clarify the role of the nasal cycle by long-term recording of nasal cycle in daily life.

Thus, the nose includes important information that reflects health state and we think that the data acquired from the nose is valuable in use for a lifelog. However, there has been no research on lifelog using the data from the nose.

In this paper, we propose a context recognition method using temperature sensors on the nostrils. The prototype system can recognize breath and nasal congestion from the temperature data in the nostril. As evaluations, we evaluated the recognition accuracy of a breath and associated behaviors, and investigated changes in nasal congestion for long-term measuring.

Note that we have already proposed the device in the poster paper [2]. The device configuration and one experiment described in Section 5.2 were almost the same as [2]. The contribution of this paper is to investigate the respiration rate recognition using temperature data in the nostrils, the recognition of some daily behaviors, and nasal congestion over a period of time.

## 2. Related Work

In existing methods for recognition of contexts and behaviors, motion sensors such as an accelerometer have been widely used, however, recent research on various sensors has enabled us to recognize a user’s psychological contexts and health. Yasufuku et al. proposed a method for stress recognition by sensing the nasal skin temperature, which changes with psychological stress [3]. As this research demonstrates, as wearable computing improves, we can gain new insights in health conditions from the nose.

As the contexts can be gathered from the nose, breathing and nasal congestion are very important for knowing user behaviors and psychological states. Breathing is one of the primary vital signs. In addition, the breathing pattern changes with age, temperature, emotion, and so on. An adult breaths with an average frequency of 15 times per minute, the ratio of inhalation to exhalation is 2:3, and the air flow in one breath is approximately 500 mL, yet breathing patterns change from cardiac disease or cerebrovascular accidents [4,5]. Moreover, breathing patterns change with emotions [6], therefore, collecting data on daily breathing can lead to improving daily life from the perspective of one’s health or monitoring one’s emotions. Various studies propose methods for detecting breathing with wearable devices. For example, Haescher et al. presented a method of respiration detection by sensing changes in an accelerometer and a gyroscope from a smartwatch or a HMD [7,8,9]. This method can detect respiration only when the user’s posture remains stable. Some consumer devices such as Hexoskin can detect breathing by measuring breathing activities through changes in thorax volume [10], however, the accuracy is not so high since the recognition based on stretch sensors is strongly affected by user movements. RS-01 can detect breathing by measuring the breath flow with nasal cannula connected to a wristband device [11]. However, we cannot usually use this device in our daily activity.

Nasal congestion is an obstruction of the nasal passage caused by the expansion of the nasal mucosa. It can be reflexively caused by a stimulator during inhalation. Moreover, nasal congestion depends on the autonomic nerves and often occurs if the parasympathetic nerve is dominant. For comparison, the sympathetic nerve is dominant when you are nervous while the parasympathetic nerve is dominant when you are relaxed. There is a biological phenomenon called a nasal cycle associated with the alternation of congestion and decongestion in right and left nostrils. The nasal cycle remains a subject of debate, which is often argued that the nasal cycle has a relationship with the activity of a cerebral hemisphere [1]. Moreover, various researchers argue that nasal congestion has a relationship with the biological state of a person. Raghuraj et al. reported that unilateral right or left nostril breathing caused by nasal congestion affects blood pressure [12]. Shannahoff-Khalsa suggested that right unilateral breathing increases heart rate compared to the left [13]. Unilateral breathing was reported to have an effect on human performance during verbal and spatial tasks [14].

Although the best way to acquire the information in the nostrils is to insert sensors into the nose, there are few research methods that acquire data in this way. In medical fields, a temperature sensor is sometimes used and is inserted into the nose for measuring breathing. However, as far as we know, this method has been used to measure breathing only temporarily. Kahana-Zweig et al. implemented a mobile device with a pressure sensor to measure the nasal cycle and measure the airflow in the nostrils while users must wear a nasal cannula [15]. There is a strong possibility to discover/clarify the role of the nasal cycle by long-term recording of nasal cycle in daily life.

However, there is no existing device for this purpose. Our proposed system can be used in daily life and it will contribute to these research areas.

## 3. System Design

### 3.1. System Structure

In this paper, a context-recognition method that uses temperature sensors in the nostrils is proposed. To extract data, small sensors are inserted in the nasal cavity, approximately two centimeters inside the nostrils. Because the diameter of a nostril is approximately one centimeter, the size of the sensors is less than one centimeter. Figure 1 illustrates the system design. The sensor in the nasal cavity is connected to a microcontroller that measures the sensor readings, which are recorded on a smartphone. This data is synced to a cloud computing system, which analyzes the gathered data and relays the classification result to the smartphone. This system enables users to obtain and verify the classification results automatically each time they extract data.

This study was approved by the research ethics committee of Kobe University (Permission number: 29-10), and was carried out according to the guidelines of the committee.

### 3.2. Selection of Sensors

We conducted pre-experiments to investigate the data from various sensors during breathing. In the pre-experiments, we collected data from the subject with a photo-reflector, temperature sensor, and humidity sensor attached to the nasal cavity. We selected these sensors because data from a photo-reflector would be affected by the shape change in breathing, data from a humidity sensor would be affected by the humidity difference between inhaling and exhaling, and data from a temperature sensor would be affected by the temperature difference between inhaling and exhaling.

The sensors were fixed in the area under the nose by taping a conductor wire in place. The subjects held their breath for a few seconds, then breathed several times. Figure 2 shows the sensors used in the pre-experiment. Figure 3 shows the results of the experiments. All sensor data shows a feature waveform of breathing. First, the photo-reflector data exhibited noise even when subjects moved their mouth or upper-lip slightly. Next, humidity sensor data exhibited delays and sometimes did not change unless subjects breathed deeply. Finally, although the temperature sensor data is affected by the air temperature, it is robust to noise and sensor size is the smallest among these sensors, thus we consider that the temperature sensor is more suited to detect breathing, and we implemented a prototype wearable device and evaluated it using a temperature sensor. On the other hand, since there is a possibility to acquire interesting contexts from other sensors, we will investigate the context awareness using different types of sensors in future.

### 3.3. Recognition Method

To recognize context, we use five feature values: the mean and variance of the raw signal and time subtraction value, and the number of crosses between the mean of signals and the raw signal. As regards the time subtraction value, we subtracted a sample from a point 20 samples ago. Because the scales of these five features are different, we normalized them so that their mean is zero and their variance is one. We use Random Forest as a classifier.

## 4. Implementation

We implemented a prototype device and a logging and monitoring application. Figure 4 shows the implemented device, and Figure 5 shows a snapshot of a person wearing the device. Figure 4 and Figure 5 are taken from the previous published paper [2]. The device consists of a pair of glasses, a case made using a 3D printer, temperature sensors, a microcontroller, and a power supply. We attached the case to where the circuit is on the glasses frame, from which the conductor wire leads to the temperature sensors in the nasal cavity along the glasses frame. As the color of the conductor wire is close to the color of the skin, it does not stand out. Moreover, as we cover the conductor wire with foundation tape, the device can appear more natural. We apply a nose mask pit in the nose and attach sensors to it so that they are not out of place. The system consists of a smartphone (Huawei Ascend G6), a microcontroller (Red Bear Lab BLE Nano), and a temperature sensor (Murata Manufacturing NXFT15XH103FA2B). The sensor data is sent from the microcontroller to the smartphone with Bluetooth Low Energy. Figure 6 shows a screenshot of the implemented application. This application shows right and left nostril temperatures in real-time and records the data in the SD card. Moreover, by syncing the SD card folder with the PC local folder using the online storage service, Dropbox, the data that we collect with the Android application is sent to the computer. We acquire recognition results from the received data using WEKA [16], a data mining software application that was developed by the University of Waikato. We send the results to the smartphone storage again by Dropbox. In this manner, users can check recognition results on their smartphones.

At this stage, there is a delay caused by data transmission between a smartphone and a PC with Dropbox. Therefore, the implementation system cannot be applied for real-time notification or alert. In the future we have plan to execute the classification and complete this system only on a smartphone.

## 5. Evaluation

We evaluate the proposed method by conducting four experiments. First, we clarify the accuracy of detecting breath using temperature data acquired from our prototype by comparing with a flow sensor. Second, we investigate whether the prototype device can recognize the workload from nasal congestion and breath. Next, we evaluate the recognition accuracy for six behaviors in one experiment (indoors) and eight behaviors over an extended period of time involving daily activities. Finally, we evaluated whether the nasal cycle can be seen from temperature data. The sampling rate for recording was 20 Hz for experiments of Section 5.1, Section 5.2 and Section 5.3 and 10 Hz for experiments of Section 5.4 and experiments of Section 5.5. The window size for calculating the feature value was 60 samples for experiments of Section 5.2 and Section 5.3 and 30 samples for experiments of Section 5.4. The window shifted by one sample. The recognition method described in Section 3.3 was used in Section 5.2, Section 5.3 and Section 5.4.

### 5.1. Evaluation of Respiration Rate Recognition

In the experiment, we evaluated the breathing detectability by comparing the respiration rate calculated by the proposed device to the ground truth. The ground truth was calculated from the data of a flow monitoring device (RS-01). Note that, this device needs a user to insert a nasal cannula to get data. Therefore, we cannot use the device in daily activity. A subject inserted a temperature sensor into one side of the nose and nasal cannula into other side of the nose. We used the number of the upward peak of the temperature data in one minute as the respiration rate for the evaluation.

The peak detection algorithm is as follows. Firstly, we applied a low-pass filter to the raw data. Next, we regarded a sample data as an upward peak if its value is larger than samples before/after it. In the calculation, we removed fake peaks. In the processing, if the detected peak value was not maximum value in a time period between detection time of the detected peak and predicted next peak, the peak was removed. We estimated the time period between a peak and predicted next peak from average seconds, which were calculated from three intervals of the past peak to peak time. Finally, we counted the detected peaks as the respiration rate. We collected the data for approximately one minute in three conditions; sitting, walking, and lying. We recruited one subject for the experiment.

Table 1 shows the respiration rates in one minute for each condition. All respiration rates detected by the proposed device were the same as the ground truth. Although the experimental conditions were limited, we confirmed that the detectability for the respiration rate using the proposed device was correct. Figure 7 shows each data measured by the flow monitoring device and the proposed device when sitting. In the comparison between the two graphs, their waveforms had a subtle difference. The temperature data graph had a small change at approximately 40 s, while the flow data graphs had a similar waveform as other points’ them. The waveforms of the temperature data graph at around 50 and 60 s were the average waveform, while the changes in the waveforms of the flow data graph were relatively small. Although the number of participants is one, the temperature changes by breathing, in principle, must be the same as other people since the changes in the waveforms depend on only breathing. Although the data of the proposed device is somewhat different from that of the flow monitoring device, the proposed device can adequately detect the respiration rates.

### 5.2. Evaluation of Workload

Because parasympathetic dominance causes nasal congestion, clearing nasal congestion by a workload can change the temperature in the nostrils. Here, we confirm how the temperature data changes by workload and discuss whether the temperature change can be used for recognition of a workload. The participants wore the prototype device and rested for 10 min in a room kept at constant temperature. They did not change their posture and location through experiment. Next, we had them perform calculation tasks for six minutes and rest for 10 minutes again. We conducted 10-fold cross-validations (the training data consisted of 90% of all samples and the remaining samples were used as test data for recognizing two contexts: high workload and rest).

We recruited five subjects for the experiments. Figure 8 shows the amplitude of the temperature for two subjects. Figure 8a is taken from the previous published paper [2]. We acquired the amplitude by calculating a maximum value among absolute values of the difference between each sample of 60 samples before and after taking a sample and an average of the 60 samples. In Figure 8, the amplitude of the temperature in the nostrils is for the congested side. The workload area in this figure is filled with yellow, showing that the amplitude for subject A increased, as shown in Figure 8a. However, only two among five subjects showed this tendency toward higher temperature amplitude during the workload experiment, while other subjects exhibited a tendency toward lower amplitudes during workload, as shown in Figure 8b. This indicates that they breathed shallowly and the airflow decreased, as indicated by the lower amplitudes of both right and left nostrils in subject E. As another change in workload, the respiration rate can be evaluated. The respiration rate of all subjects increased during the workload experiments, which affected the feature value of the number of crosses between the signal mean and the raw signal. Table 2 shows precision, recall, and recognition rate for each participant and the averages for all participants. From the above, the recognition rate for all subjects was over 90%. Table 2 is taken from the previous published paper [2]. Although the recognition accuracy rate was high, the classification result has a bias and the experiment result was not replicable. Therefore, we need to collect more data in this experiment and regenerate and evaluate the classification model in order to implement an actual application.

### 5.3. Evaluation of Behavior Recognition

We investigated whether the proposed system can recognize behaviors that can change the temperature in the nostrils or affect breath. We evaluated the recognition accuracy of six behaviors: drinking, remaining rest, eating, walking, laughing, and vocalizing. The participants performed three sets of these behaviors and performed each behavior for 30 s. We conducted 3-fold cross-validations. As the post processing, we apply majority processing to acquired contexts. Concretely, our method re-label the context for a sample as a label that appears most often before/after 30 samples. We recruited four participants. Table 3 shows the confusion matrix of all subjects. A confusion matrix is a table that indicates the number of samples where a behavior is recognized. As Table 3 shows, the average recognition rate was 54%, the highest recall rate of the behavior walking was 76%. The walking category reflects subjects walking on a treadmill at 6 kph, which forced subjects to breathe hard and highlighted the variance value. These tests resulted in high recall rates. The lowest recall rate was obtained for the behavior laughing at 29%, which was often recognized as volcalizing incorrectly. This is because an irregular temperature fall in laughing was similar to a temperature fall in volcalizing when inhaling. As Table 4 shows, average precision rates of subjects were similar, yet the lower recognition rate behaviors were different among subjects. This is because the temperature in the nostrils changed entirely even when subjects lightly breathed and it depended on the specific individual’s habits. Therefore, we will need to use an individual’s data as training data in a practical application. The reason why the recognition rate was generally low was that the temperature change depended on typical breathing, and a global feature was difficult to capture. Therefore, as future work, we plan to investigate if a particular behavior will produce a characteristic temperature change. Moreover, we plan to use waveform matching such as DTW to support this effort.

### 5.4. Evaluation Using Natural Data in Daily Activities

We investigated whether the proposed system can recognize behaviors in daily life based on temperature changes by collecting data over an extended period of time. We conducted the experiment for seven hours for one day in August, 2017. We evaluated eight behaviors, adding yawning and sneezing to the six behaviors from the previous section. We conducted 10-fold cross-validations. We labeled the data as behaviors based on video recorded during the experiment. We labeled samples for one period where the subject performed a behavior except at rest as the behavior and other periods as at rest. Moreover, to reduce the number of samples that were labeled as at rest to be similar to other behaviors, we subsampled them every 20 samples. We recruited one participant. Table 5 shows the resulting confusion matrix. The average recognition rate of all behaviors was 86%. All behaviors were often labeled incorrectly as walking and vocalizing, and the precision rates of these two behaviors were lower. Recall and precision rates of at rest were the lowest. This is because the temperature change largely depended on typical breaths and the significant information was lost by subsampling. The recognition rate was higher than the recognition rate of the previous section’s experiment because we collected the training data and test data from a series of experimental data without taking off and on the device, as in the experiment of the evaluation of workload. However, in applications such as lifelog services, we need to use the training data collected in advance. Therefore, we plan to conduct the experiment again with this change. From this experiment, we already know which points to consider. First, one is the point when a difference between indoor and outdoor temperatures occurs. We can solve this problem by using an additional temperature sensor for measuring the outside temperature. We can further recognize more contexts by using this change. For example, by measuring the temperature change when entering and exiting a room, the system can classify whether the user is indoors or outdoors and can more correctly recognize the behaviors and contexts that we often perform under each environment. Next, another point that we need to consider is the problem that the temperature changes in a few hours period depending on the change in nasal airflow in the nasal cycle. Although right or left nasal airflow changes periodically, the sum of the right and left nasal airflows is essentially constant. Therefore, we should be able to collect constant data by summation of the temperatures in the right and left nostrils.

### 5.5. Investigating Nasal Congestion over a Period of Time

We investigated whether we could recognize the nasal cycle from raw temperature data. We recruited four participants. They wore the prototype device for from six to seven hours and carried an Android smartphone for collecting the data. We did not restrict their activities during experiments except to prohibit them from lying down because lateral recumbency causes nasal congestion in the nasal downside [17]. After we calculated the amplitude of temperature using the same method as the workload recognition experiments, we processed the data to compare the right and left data. First, we averaged the data in 300 samples. Next, we normalized the right and left data so that the maximum value was one and the minimum value was zero to correct for the amplitude differences between the right and left data. Figure 9 shows the temperature amplitudes after being processed for the four subjects. In the experiment of subject D, the battery of the prototype device died and the measurement stopped. As Figure 9a shows, we can see a cycle in subject A clearly. Subject A’s cycle is approximately 100 min, where one side’s value is greater than the other side’s value. Regarding the other subjects, we cannot see a clear cycle in the graphs, however, we can see an alternation of the side whose amplitude is greater. For example, in the result of subject C the amplitude of the right nostril was greater from zero hours to approximately one and one-half hours, however, after that time, the amplitude of the left side was greater. Similarly, in the result of subject B the amplitude of the left nostril was greater from zero hours to approximately two hours, however, afterward, the amplitude of the right side was greater. In the result of subject D, although we can see that the amplitude of the right nostril was much greater than the amplitude of the left nostril, we cannot see a cycle or an alternation of the side with greater amplitude. From the above, although the system can detect the nasal cycle, the results of the experiment were different between subjects and we need to collect more data for further analysis. If we can determine a clear relationship between the nasal cycle and other biological information, we can apply the proposed system to the medical field or perhaps provide a novel lifelog service. In this long-term experiment, the temperature sensors were wet by nasal mucus, yet the sensor data did not appear to be significantly affected by this. However, we need to confirm how much nasal mucus affects the sensor data. We received feedback that the device interferes when a user wishes to blow their nose. Accordingly, we need to improve the design of the device.

### 5.6. Possibility as a User Interface

We confirmed that the temperature sensor on the proposed device showed high sensitivity and was significantly affected by not only simple breaths but various behaviors. Therefore, we can apply the proposed device to an input user interface with which we operate a computer with unusual breathing. Because this user interface senses only slight breathing, it can provide high confidentiality and operation without motion. Further, if we can combine this interface with an HMD, we can browse information and operate the display using only glasses. As a specific operation, we can perform a simple input such as “Yes or No” by controlling inhaling and exhaling. In addition, we can input characters by using inhaling and exhaling as Morse code. We plan to explore implementing a user interface as described above.

## 6. Conclusions

In this paper, we proposed a context recognition method using temperature sensors in the nostrils. In the experiment, we selected the best sensor suitable for detecting breath from among a photo-reflector, humidity sensor, and temperature sensor. We confirmed that a temperature sensor was more suitable for detecting breath correctly. We implemented a prototype device using temperature sensors and evaluated the recognition of breath, workload, and behaviors. In the recognition of breath, we could see that the proposed system can recognize breath correctly, although the temperature data behaved differently than the baseline flow data. We confirmed that the change in the temperature data caused by the workload was different between subjects and did not confirm that nasal congestion changed with workload. However, the recognition rate was 96.4% and it was primarily attributed to changes in respiration rate. The proposed system could recognize six behaviors: drinking, keeping the rest, eating, walking, laughing, and vocalizing at an average 54% accuracy and eight behaviors: drinking, keeping the rest, eating, walking, laughing, vocalizing, yawning, and sneezing at an 86% accuracy rate on average. As future work, we plan to improve the device regarding sanitation and wear comfort. In this paper, we conducted the experiment under certain separate circumstances, therefore, we need to confirm whether the proposed system can recognize psychological contexts and behaviors from a single data source. We plan to consider the difference between indoor and outdoor temperatures and their influence on the nasal cycle, and implement the application.

## Figures and Tables

**Figure 1 sensors-19-01528-f001:**
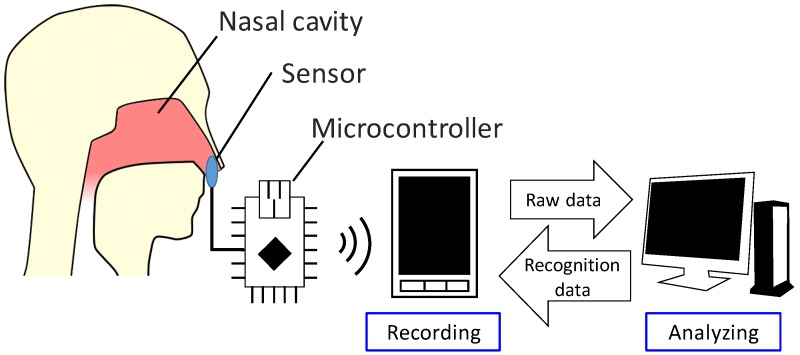
Hardware structure.

**Figure 2 sensors-19-01528-f002:**
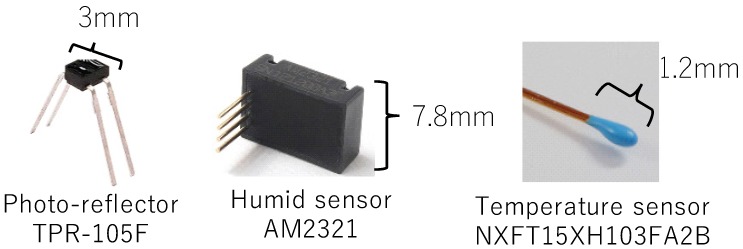
Sensors used in the pre-experiment.

**Figure 3 sensors-19-01528-f003:**
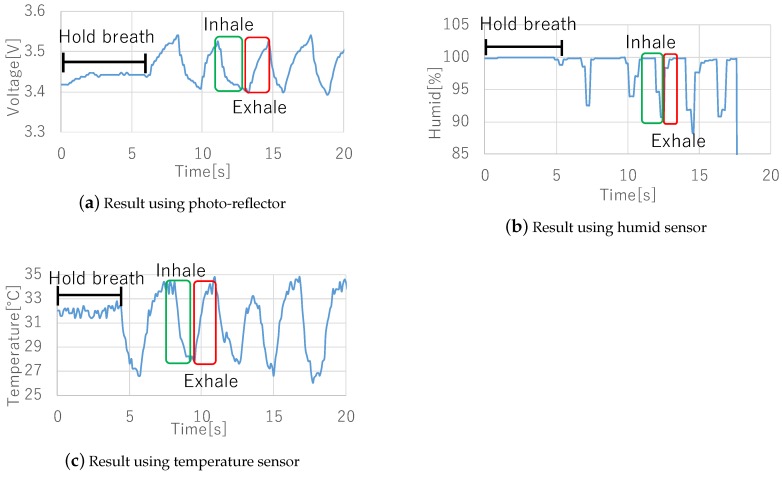
Result of three sensors.

**Figure 4 sensors-19-01528-f004:**
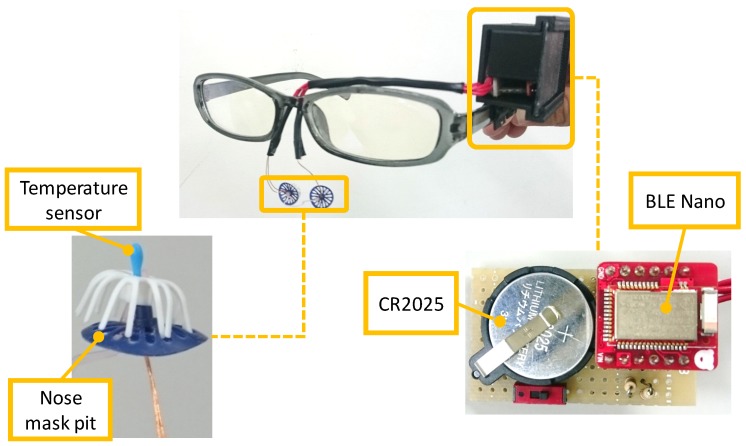
Prototype device [2].

**Figure 5 sensors-19-01528-f005:**
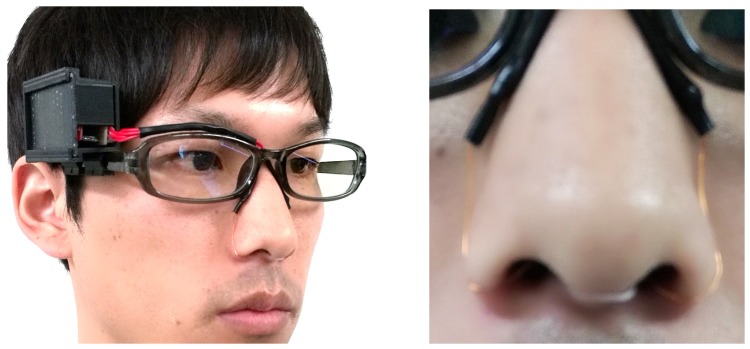
A user wearing the prototype [2].

**Figure 6 sensors-19-01528-f006:**
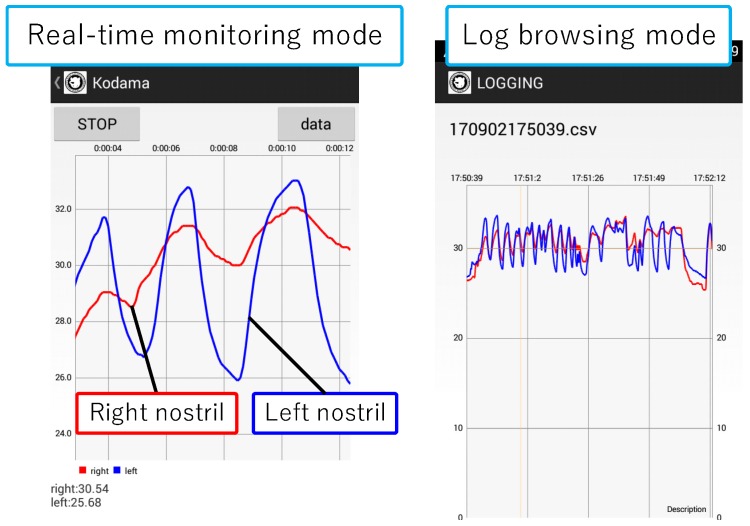
Screen of the Android application.

**Figure 7 sensors-19-01528-f007:**
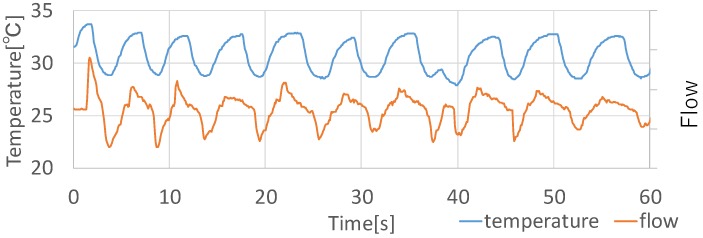
Results of airflow and temperature.

**Figure 8 sensors-19-01528-f008:**
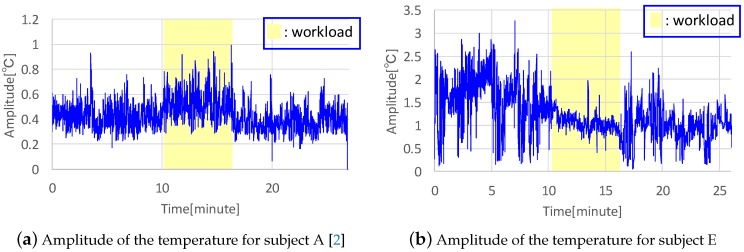
Results of two subjects’ amplitudes.

**Figure 9 sensors-19-01528-f009:**
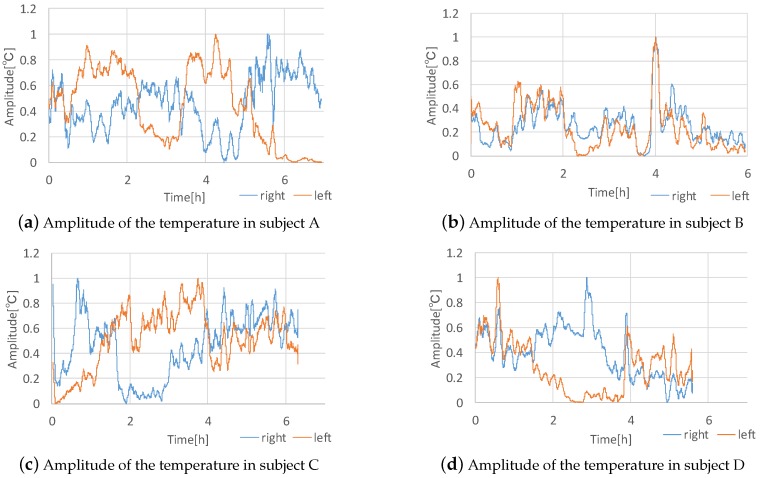
Results of subjects’ temperature amplitudes.

**Table 1 sensors-19-01528-t001:** Respiration rates in one minute.

	Proposed Device	Ground Truth
Sitting	10	10
Walking	10	10
Lying	12	12

**Table 2 sensors-19-01528-t002:** Workload recognition results [2].

	Precision [%]	Recall [%]	Accuracy [%]
A	96.5	96.5	97.5
B	94.7	89.0	94.3
C	99.5	99.5	99.6
D	94.7	96.0	97.1
E	92.2	89.2	93.5
average	95.7	94.0	96.4

**Table 3 sensors-19-01528-t003:** Confusion matrix of all subjects.

	Prediction						Recall
	1	2	3	4	5	6	
1: drinking	2867	1363	430	393	959	476	0.44
2: eating	1773	3040	430	544	496	541	0.45
3: walking	514	189	4957	44	567	218	0.76
4: laughing	1086	879	8	1893	900	1724	0.29
5: rest	194	426	937	596	4289	74	0.66
6: vocalizing	387	436	0	1455	78	4133	0.64
Precision	0.41	0.44	0.77	0.42	0.59	0.58	0.54

**Table 4 sensors-19-01528-t004:** Precision of each behavior for all subjects.

	Subject				Average
	A	B	C	D	
1: drinking	0.39	0.44	0.52	0.20	0.39
2: eating	0.40	0.55	0.41	0.49	0.46
3: walking	0.45	0.90	0.99	0.89	0.81
4: laughing	0.62	0.48	0.28	0.36	0.44
5: rest	0.33	0.81	0.54	0.59	0.57
6: vocalizing	0.65	0.75	0.35	0.64	0.60
Average	0.47	0.66	0.51	0.53	

**Table 5 sensors-19-01528-t005:** Confusion matrix for eight behaviors.

	Prediction								Recall
	1	2	3	4	5	6	7	8	
1: drinking	912	8	54	6	50	128	4	0	0.78
2: eating	5	8619	284	7	273	531	7	0	0.89
3: walking	10	54	20,622	7	619	713	15	3	0.94
4: laughing	5	7	35	892	12	135	0	0	0.82
5: rest	27	27	1002	28	6044	2049	93	5	0.63
6: vocalizing	17	236	1131	31	995	20,553	26	3	0.89
7: yawning	0	11	21	2	42	124	1309	0	0.87
8: sneezing	0	0	5	1	5	24	1	116	0.76
Precision	0.93	0.92	0.89	0.92	0.75	0.85	0.90	0.91	0.86

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
