# Peer review of "Evaluation on Context Recognition Using Temperature Sensors in the Nostrils"

_sensors, 2019, doi:10.3390/s19071528_

Reviewer 1 Report

In this article, the authors proposed a context recognition method using temperature sensors attached to the nasal cavity. They have developed a prototype system that acquires temperature data from human nostrils using small temperature sensors. Experimental evaluations conducted by the authors show that the newly proposed system can recognise breathing to 96.4% accuracy, 6 human behaviours to an accuracy of 54% and 8 behaviours of daily life to an accuracy of 86%.

This is an excellently, well written and well-structured article with great potentials.  One of the unique contributions of the article is its capability to detect nasal congestions, with high potentials for application in medical field.  In addition, the article proposed a novel device based on temperature sensor unlike the existing proposals that use pressure sensors, gyroscopes or accelerometers for identifying nasal congestions. Strength of the proposal is the prototyping, experimentation and implementation.

I have made the following few observations:

The data collected is read through a microcontroller and recorded on a smartphone which transmits the data to the cloud for classification. However, the authors have not evaluated or informed us if there is any delay between transmission of the data to cloud and when the classification result is obtained by the smartphone. Is it not possible to execute the classification on the smartphone?

In section 4.1, evaluation of respiration rate recognition, the authors collected data from only one subject, why? What is the impact of this on the generalisation of the classification model?

The last sentence in section 4.1(line 165&166) (Although the data of the proposed device have somewhat different from the data of the flow monitoring device…) should be reviewed and re-written.

In section 4.2, why use 90% of the data for training? This might biase the classification results. Any evidence that this did not happen?

In section 4.6, Yesor No, should be "Yes or No"

In the related work section, on line 56, revise this phrase …since the recognition based on stretch sensors is strongly affect from user…

Finally, no reference for WEKA in the reference list!

Author Response

Point 1: The data collected is read through a microcontroller and recorded on a smartphone which transmits the data to the cloud for classification. However, the authors have not evaluated or informed us if there is any delay between transmission of the data to cloud and when the classification result is obtained by the smartphone. Is it not possible to execute the classification on the smartphone?

Response 1: As you pointed out, there is a delay between the transmission of the data between a smartphone and a PC. Therefore, the implementation system cannot be applied for real-time notification or alert. Although it is possible to execute the classification on the smartphone, we executed the classification on the PC because implemented android application was unstable and a long time use of the application is not sure. In the future, we have plan to execute the classification and complete this system only on a smartphone.

As revised points, we added the second paragraph in Section 4 and added an explanation that mentioned above there.

Point 2: In section 4.1, the evaluation of respiration rate recognition, the authors collected data from only one subject, why? What is the impact of this on the generalization of the classification model?

Response 2: We apologize for making you confused, but we did not generate the classification model in section 4.1, evaluation of respiration rate recognition. In section 4.1, we only conducted the peak detection from the raw data and calculated the number of peak as the respiration rate. We generated the classification model in section 4.2, section 4.3, section 4.4.

  As revised points, we added an explanation about the evaluations where we generated the classification model in line 147, first paragraph, section 5.

As you pointed out, the number of participants is small. However, we do not think that in principle, the waveforms of breathing are quite different between participants and the evaluation is adequate.

  As revised points, we added the explanation in line 172.

Point 3: The last sentence in section 4.1(line 165&166) (Although the data of the proposed device have somewhat different from the data of the flow monitoring device…) should be reviewed and re-written.

Response 3: We re-wrote the sentence, Although the data of the proposed device have somewhat different from the data of the flow monitoring device, the device can adequately detect the respiration rates” into the sentence, “Although the data of the proposed device is somewhat different from that of the flow monitoring device, the proposed device can adequately detect the respiration rates” in line 173-174.

Point 4: In section 4.2, why use 90% of the data for training? This might biase the classification results. Any evidence that this did not happen?

Response 4: As you pointed out, the classification result has a bias and the experiment result was not replicable. Therefore, we need to collect more data in this experiment and regenerate and evaluate the classification model in order to implement an actual application.

  As revised points, we added the explanation in line 199, second paragraph, section 5.2.and added an explanation about the purpose of evaluation in line 178, first paragraph, section 5.2.

Point 5: In section 4.6, Yesor No, should be "Yes or No"

Response 5: We re-wrote Yesor No into "Yes or No" in line 285.

Point 6: In the related work section, on line 56, revise this phrase …since the recognition based on stretch sensors is strongly affect from user…

Response 6: We re-wrote the phrase, since the recognition based on stretch sensors is strongly affect from user…  into the phrase, since the recognition based on stretch sensors is strongly affected by user… in line 58.

Point 7: Finally, no reference for WEKA in the reference list!

Response 7: We added the reference for WEKA in line 350 and the citation in line 131.

Reviewer 2 Report

Please take a look in the following suggestions:

- Introduction must be section 1, not 0;

- I think the authors must discuss more in the introduction about:

  -- the main differences related to other work;

  -- the increase made in this paper related to the previous published paper (table 2 is the SAME as the published paper);

  -- the research problem.

- In the section 1 I could not find a direct related work. Is there any sensor described in the literatura with similar propose ?

- I think the authors can improve the description of the structure of section 2.1. 

Author Response

Point 1: Introduction must be section 1, not 0

Response 1: We would like to thank for your advice. We re-wrote all section number.

Point 2: I think the authors must discuss more in the introduction about:

-- the main differences related to other work;

Response 2: The main differences related to other work is that we collect the data from the nostrils. There has been no research on lifelog using the data from the nose.

  As revised points, we added the explanation in line 30, paragraph 2, section 1.

 Point 3: I think the authors must discuss more in the introduction about:

-- the increase made in this paper related to the previous published paper (table 2 is the SAME as the published paper);

Response 3: I apologize, but I did not understand what we should discuss about the increase made in this paper related to the previous published paper. Although in the last paragraph, section 1 we described the difference from the previous published paper, is it not enough?

Point 4: I think the authors must discuss more in the introduction about:

-- the research problem.

Response 4: We think that the research problem of our research is that nasal congestion bothers many people, however, its causes are complicated and remain unknown. The problem should be solved by logging the state of the nose for a long time.

  As revised points, we added the explanation about these research problems in line 29, paragraph 2, Section 1.

Point 5: In the section 1 I could not find a direct related work. Is there any sensor described in the literatura with similar propose ?

Response 5: There has been no research on lifelog using the data from the nose. As a method of insert a device into the nose, in medical fields, a temperature sensor is sometimes used and is inserted into the nose for measuring breathing. However, as far as we know, this method has been used to measure breathing only temporarily.

  As revised points, we added the explanation about a direct related work in line 73, third paragraph, section 2.

Point 6: I think the authors can improve the description of the structure of section 2.1.

Response 6: We have applied this section to English proofreading. As soon as it is revised, we will resubmit again.

Round  2

Reviewer 2 Report

The paper can be accepted in current form.